# Stationary Activations for Uncertainty Calibration in Deep Learning

**Lassi Meronen**
Aalto University / Saab Finland Oy
Espoo, Finland
lassi.meronen@aalto.fi

**Christabella Irwanto**
Aalto University
Espoo, Finland
christabella.irwanto@aalto.fi

**Arno Solin**
Aalto University
Espoo, Finland
arno.solin@aalto.fi

## Abstract

We introduce a new family of non-linear neural network activation functions that mimic the properties induced by the widely-used Matérn family of kernels in Gaussian process (GP) models. This class spans a range of locally stationary models of various degrees of mean-square differentiability. We show an explicit link to the corresponding GP models in the case that the network consists of one infinitely wide hidden layer. In the limit of infinite smoothness the Matérn family results in the RBF kernel, and in this case we recover RBF activations. Matérn activation functions result in similar appealing properties to their counterparts in GP models, and we demonstrate that the local stationarity property together with limited mean-square differentiability shows both good performance and uncertainty calibration in Bayesian deep learning tasks. In particular, local stationarity helps calibrate out-of-distribution (OOD) uncertainty. We demonstrate these properties on classification and regression benchmarks and a radar emitter classification task.

## 1 Introduction

Deep feedforward neural networks (see, *e.g.*, [26, 28]) have become an essential component of modern machine learning. Their black box nature results in a lack of interpretability, an issue that has been tackled recently from many directions, one of which is the study of random (untrained) networks in order to examine what prior assumptions they impose over functions. By assuming a probability distribution on the network parameters, a distribution is induced from the inputs to the outputs of the network. While we typically want networks to have high modelling capability (or *flexibility*), large networks can be hard to analyse directly. This difficulty motivates instead the study of the limiting behaviour of the networks, which can provide new insight and better interpretation. This is also of interest for *Bayesian deep learning*, where the concept of assigning priors on neural networks only makes sense if the effects of prior assumptions can be understood.

Along the line of studying random networks, Neal [47] showed that under certain assumptions, random neural networks with one hidden layer converge to a Gaussian process (GP, [53]) in the limit of infinite width. Since then, explicit links between common neural network activation functions and GP covariance (kernel) functions have been shown (*e.g.*, ERF and RBF activations by [68] and ReLU and step activations by [8], see Fig. 1). Recently, this equivalence was extended to deep networks [14, 39]. Gal and Ghahramani [21] leveraged the connection for approximate variational inference in neural networks. The call for more work on deep net priors (*e.g.*, [19, 49, 62]) is also motivated by empirical findings [67], and we recognize two separate problems in deep learning: *(i)* designing network architectures that support our domain knowledge of the modelling task (prior assumptions), and *(ii)* performing inference and learning with the model. Recent interest in Bayesian deep learning has been in *(ii)* (see, *e.g.*, [4, 35, 41, 55]), while we tackle *(i)*, and choose to apply a simple and conservative inference method—we resort to Monte Carlo (MC) dropout in our experiments.

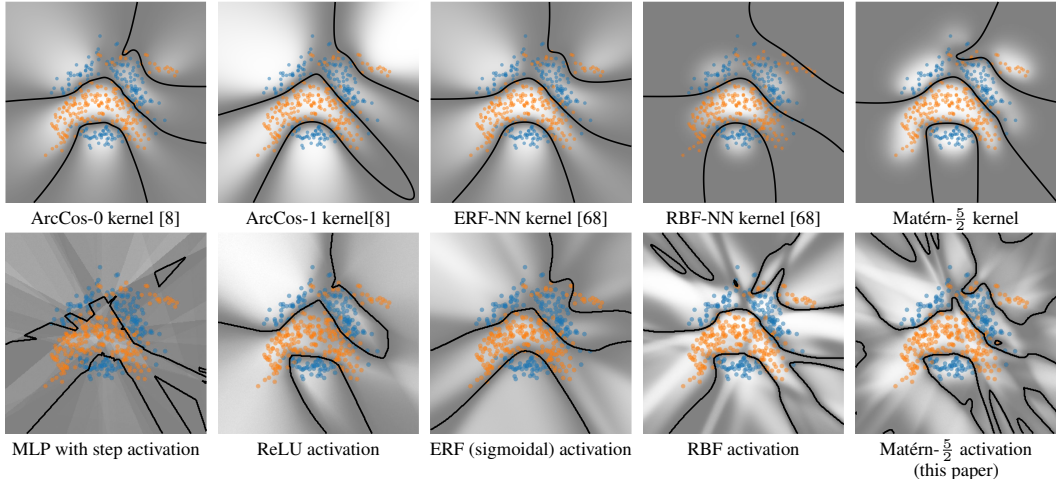

| | | | | |
|---|---|---|---|---|
| ArcCos-0 kernel [8] | ArcCos-1 kernel[8] | ERF-NN kernel [68] | RBF-NN kernel [68] | Matérn-$\frac{5}{2}$ kernel |
| MLP with step activation | ReLU activation | ERF (sigmoidal) activation | RBF activation | Matérn-$\frac{5}{2}$ activation (this paper) |

Figure 1: Illustrative comparisons on the *Banana* classification data set. The top row shows decision boundaries and marginal predictive variance (low ▭ high) for various GP priors (infinite-width networks), and the bottom row shows finite-width MLP neural network results (one hidden layer, 50 nodes, MC dropout) with the network activation function matching the GP prior on the top row.

Orthogonal to deep learning, in support vector machines [10], kernel methods [36], spatial statistics [11], and GP models, the focus is on the choice and crafting of a *kernel* (covariance/covariogram function). The kernel captures prior assumptions related to the model functions such as continuity, differentiability, periodicity, invariances, *etc.* Stationarity (translation invariance of the kernel) is often a sought-after property in these models as it induces the behaviour of functions reverting back to the prior outside informative regions in the problem domain (decision boundaries/data samples). Arguably the most used GP kernel is the Matérn class [42, 53], which features stationary kernels with continuous sample functions of various degree of smoothness. This class has the RBF (squared exponential or Gaussian) and the exponential (Ornstein–Uhlenbeck) covariance functions as limiting cases of infinite smoothness and non-differentiable sample paths, respectively (see, *e.g.*, [53]).

Previous approaches have sought to map various activation functions to their kernel counterparts. We take the opposite approach by introducing a new family of non-linear neural network activation functions that mimic the widely-used Matérn family in GP models. The derivation is made possible by rekindling the link between classical control theory and neural networks. The class we propose spans a range of locally stationary models of various degrees of mean-square differentiability. We show an explicit link to the corresponding GP models in the case of one infinitely wide hidden layer. These activation functions result in similar appealing properties to their counterparts in GP models, and we demonstrate that the local stationarity property together with limited mean-square differentiability shows good performance and uncertainty quantification in Bayesian deep learning tasks. In particular, the local stationarity can help in tackling overconfidence in out-of-distribution detection.

## 1.1 Related Work

We naturally desire predictive models to know what they do not know: that is, to have well-calibrated out-of-sample or *out-of-distribution (OOD)* detection. This is often of vital concern for safety-critical applications [2]. However, overly confident OOD predictions is a well-recognized problem in state-of-the-art discriminative neural networks [25, 31, 40, 48] and deep generative models [9, 32, 46, 54]. To clarify our terminology, we consider 'OOD detection' to be a broad problem that includes 'anomaly detection'/'outlier detection' [9, 32]), 'open-set recognition' [5, 15], and robustness to 'domain shift', 'data set shift', and 'covariate shift' [6, 59] or 'rubbish class' [25] and 'fooling' [48] examples. OOD data can refer to unseen classes from the same data set (*e.g.*, unknown bacteria [54] or fraudulent credit card transactions [9]), or a different data set (*e.g.*, NOTMNIST from MNIST [37, 55], SVHN from CIFAR-10 [46]). Data can also be OOD by construction: 'rubbish class' or 'fooling' examples are synthetic inputs, *e.g.*, random noise images [48]. OOD inputs are generally *far away* from training data, whereas an arguably related problem arises from minor perturbations from the training data, such as common corruptions [30] and adversarial examples [63].

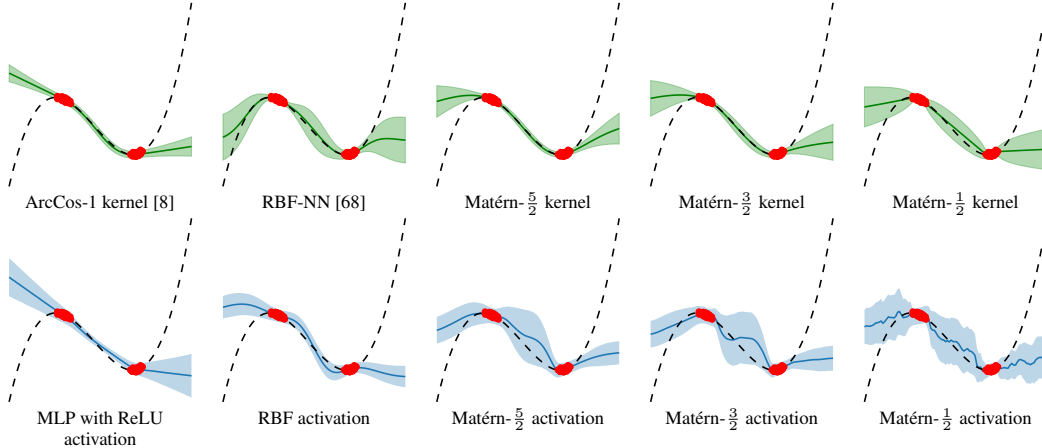

Figure 2: Illustrative examples of 1D regression tasks (shading covering two sigma) with GP models on the top row and neural networks (50 hidden units, MC dropout) on the bottom. Our proposed models capture similar behaviour as their GP counterparts, with 'stiffness' reducing from left to right. The predictive variance scale for the neural network models is strongly affected by the choice of hyperparameters, such as the dropout rate, and the focus of this example is showing how the variance around the training data relates to the variance away from training points.

For many classifier neural networks, part of the problem is intrinsic to their very architecture. Good-fellow *et al.* [25] attributed overconfidence on far-away 'rubbish' examples to the linear behaviour of ReLU networks in high dimensions, while RBF networks were shown to be immune. This was later formalized, in that all networks with piecewise affine activations (*e.g.*, ReLU, leaky ReLU) lead to continuous piecewise affine classifiers [3, 12], which produce predictions with arbitrarily high confidences on far-away data, whereas RBF networks have almost-uniform confidence [29]. While many OOD detection methods rely on specific assumptions or problem formulations (such as knowledge of the OOD distribution [15, 29], or augmenting classifier with a reject class (see [15] background section for examples)), in our work we directly address the neural network prior itself, which is largely governed by its *activation functions*.

In contrast to neural networks, GPs with stationary kernels do not overconfidently extrapolate far from the training data, much like RBF networks. However, they have limited representational power, but would perhaps be more widely applied if not for their associated computational complexity. As such, we use the previously mentioned GP-neural network correspondence to incorporate the desirable OOD behaviour of Matérn kernel GPs into neural networks, via a new class of activation functions. Related work addressing the modelling problem in deep learning has similarly revolved around designing functional priors [19, 62], such as through BNN architectural choices [49]. However, the focus has not been on improving OOD behaviour, with the exception of noise contrastive priors (NCPs, [27]) where, however, generating noisy samples at data boundaries can be hard in practice.

Our approach of studying stationary kernels through their spectral density is related to a wide range of previous work in Gaussian process models. This duality has given rise to leveraging Fourier features (see, *e.g.*, [34, 38, 51, 60]) by projecting the GP problem on a set of harmonic basis functions. While we share the idea of using the Fourier duality, the resulting model is spanned by different basis functions. Conventional sinusoidal Fourier features enforce (global) stationarity with the approximation based on Eq. (5), while our approach is locally stationary as defined by Eq. (2). This same global vs. local difference is also shared between this work and the SIREN activations of [58].

## 2 Random Feedforward Networks and Gaussian Processes

Gaussian process (GP, [53]) models admit the form of a GP prior $f(\mathbf{x}) \sim \mathcal{GP}(\mu(\mathbf{x}), \kappa(\mathbf{x}, \mathbf{x}'))$ and a likelihood (observation) model $\mathbf{y} \mid \mathbf{f} \sim \prod_{i=1}^{n} p(y_i \mid f(\mathbf{x}_i))$, where the data $\mathcal{D} = \{(\mathbf{x}_i, y_i)\}_{i=1}^{n}$ are input–output pairs, $\mu(\mathbf{x})$ the mean function, and $\kappa(\mathbf{x}, \mathbf{x}')$ the covariance function of the GP prior. This probabilistic machine learning paradigm covers many standard modelling problems, including regression and classification tasks, where GPs not only predict well but also enable uncertainty

estimation and model selection via the marginal likelihood. The Gaussian process is completely specified by its mean and covariance function, which encapsulate the assumptions about the sample processes $f$ (such as continuity, differentiability, periodicity, *etc.*):

$$\mu(\mathbf{x}) := \mathrm{E}[f(\mathbf{x})] \qquad \text{and} \qquad \kappa(\mathbf{x}, \mathbf{x}') := \mathrm{E}[(f(\mathbf{x}) - \mu(\mathbf{x}))(f(\mathbf{x}) - \mu(\mathbf{x}))^*]. \qquad (1)$$

Without loss of generality, we limit our interest to zero-mean ($\mu(\mathbf{x}) := 0$) GP priors. In practice implementations work with an $n \times n$ Gram (covariance) matrix $\mathbf{K}$ with $\kappa(\mathbf{x}_i, \mathbf{x}_j)$ as the $ij^{\text{th}}$ entry for $\forall \mathbf{x} \in \mathcal{D}$. This gives rise to a prohibitive cubic scaling $\mathcal{O}(n^3)$ in the number of data due to associated matrix inversions, as well as the non-parametric nature of GPs, where the number of parameters in the model is not fixed, but rather spanned by the number of data points.

The link between feedforward neural networks (NN) and GPs is generally well understood [14]. Neal [47] showed that a random (untrained) single-layer network converges to a GP in the limit of infinite width. Let $\sigma(\cdot)$ be some non-linear (activation) function such as the ReLU or sigmoid, and $\mathbf{w}$ and $b$ be the network weights and biases. We can define the associated kernel for the infinite-width network (under assumptions on Gaussian weights) to be formulated in terms of [47]

$$\kappa(\mathbf{x}, \mathbf{x}') = \int p(\mathbf{w}) \, p(b) \, \sigma(\mathbf{w}^{\mathsf{T}} \mathbf{x} + b) \, \sigma(\mathbf{w}^{\mathsf{T}} \mathbf{x}' + b) \, \mathrm{d}\mathbf{w} \, \mathrm{d}b, \qquad (2)$$

where $p(\mathbf{w})$ is a multivariate distribution and $p(b)$ is a univariate Gaussian. In the absence of a closed-form solution, a Monte Carlo approximation to the above integral arrives at a random or rank-deficient (see [21]) formulation for the covariance function

$$\hat{\kappa}(\mathbf{x}, \mathbf{x}') = \frac{1}{K} \sum_{k=1}^{K} \sigma(\mathbf{w}^{\mathsf{T}} \mathbf{x} + b) \, \sigma(\mathbf{w}^{\mathsf{T}} \mathbf{x}' + b) \qquad (3)$$

with $\mathbf{w} \sim p(\mathbf{w})$ and $b \sim p(b)$, and $K$ having the interpretation of the number of hidden units in a single hidden layer NN approximation. This representation has links to *Mercer's theorem* [44] which states that any positive-definite kernel can be represented as the inner product between a fixed set of features, evaluated at $\mathbf{x}$ and $\mathbf{x}'$: $\kappa(\mathbf{x}, \mathbf{x}') = \mathbf{h}(\mathbf{x})^{\mathsf{T}} \mathbf{h}(\mathbf{x}')$. From the formulation in Eq. (2) it is easy to see that for a given $\sigma(\cdot)$ the corresponding kernel can be recovered by solving this integral either by sampling as in Eq. (3) or in closed form as has been done, *e.g.*, for the RBF and ERF [68], ReLU and step [8], leaky ReLU [65], and cosine activations [49] (*cf.* Fig. 1). However, solving the *inverse problem* of recovering $\sigma(\cdot)$ given $\kappa(\cdot, \cdot)$ is typically harder and has received less attention.

## 3 Methods

We derive activation functions $\sigma(\cdot)$ (non-coincidentally called *transfer functions* in 1980s and '90s neural networks literature) resulting in random networks that capture the behaviour induced by the Matérn class [42, 53]. We start from a fully stationary setting, which we then relax to local stationarity in Sec. 3.2 in order to match the implicit input density assumption in random networks (giving our main result). We conclude this section by providing an alternative view through functional analysis.

### 3.1 Transfer Function Approach to Activation

A *stationary* (homogeneous) covariance function is invariant to translations of the input space. This means that the covariance structure of the model functions $f(\mathbf{x})$ is the same regardless of the absolute position in the input space, and thus the covariance function can be parameterized as $\kappa(\mathbf{x}, \mathbf{x}') \triangleq \kappa(\mathbf{x} - \mathbf{x}') = \kappa(\mathbf{r})$. For stationary GP priors, the covariance function can be written equivalently in terms of its spectral density function. This results from *Bochner's theorem* (see, *e.g.*, [1, 13]) which states a bounded continuous positive definite function $\kappa(\mathbf{r})$ can be represented as

$$\kappa(\mathbf{r}) = \frac{1}{(2\pi)^d} \int \exp\left(\mathrm{i}\, \boldsymbol{\omega}^{\mathsf{T}} \mathbf{r}\right) \mu(\mathrm{d}\boldsymbol{\omega}), \qquad (4)$$

where $\mu$ is a positive measure and $\mathbf{r} \in \mathbb{R}^d$. If the measure $\mu(\boldsymbol{\omega})$ has a density, it is called the *spectral density* $S(\boldsymbol{\omega})$ corresponding to the covariance function $\kappa(\mathbf{r})$. This is the Fourier duality of covariance and spectral density, which is known as the *Wiener–Khinchin theorem* (see, *e.g.*, [53]):

$$\kappa(\mathbf{r}) = \frac{1}{(2\pi)^d} \int S(\boldsymbol{\omega}) \exp(\mathrm{i}\, \boldsymbol{\omega}^{\mathsf{T}} \mathbf{r}) \, \mathrm{d}\boldsymbol{\omega} \quad \text{and} \quad S(\boldsymbol{\omega}) = \int \kappa(\mathbf{r}) \exp(-\mathrm{i}\, \boldsymbol{\omega}^{\mathsf{T}} \mathbf{r}) \, \mathrm{d}\mathbf{r}. \qquad (5)$$

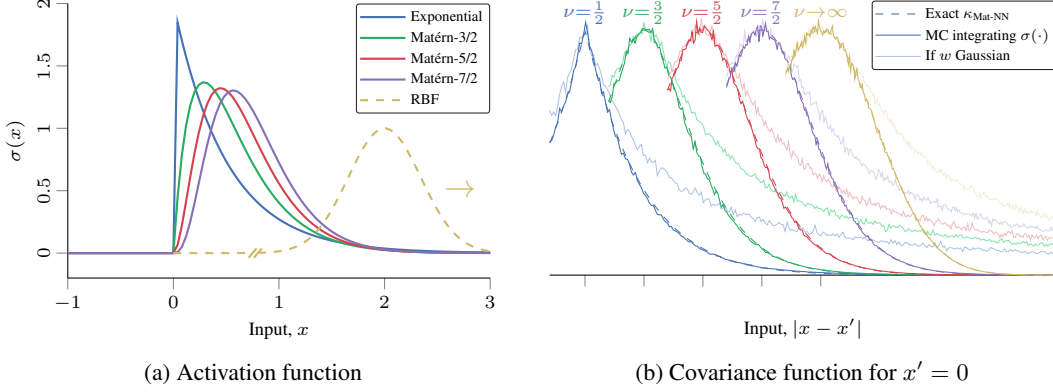

(a) Activation function      (b) Covariance function for $x' = 0$

Figure 3: (a) Activation functions $\sigma(\cdot)$ corresponding to the Matérn kernel class for various typical degrees of smoothness $\nu$. (b) Corresponding covariance functions calculated by MC integration and compared to their exact dashed counterparts (peaks shifted for clarity). Light curves are the same under Gaussian weights instead of binary white weights. When $\nu \to \infty$, we recover the RBF-NN [68].

If $d > 1$, these identities show that if the covariance function is *isotropic*—that is, it only depends on the Euclidean norm $\|\mathbf{r}\|$ such that $\kappa(\mathbf{r}) \triangleq \kappa(\|\mathbf{r}\|)$ (thus being invariant to all rigid motions of the input space)—then the spectral density will also only depend on the norm of the dual input variable $\boldsymbol{\omega}$. In one-dimension these definitions coincide, and in the following we restrict our interest to 1D projections in the spirit of Eq. (2). From the above, we can also define the corresponding (power) spectral density of the process, which is the square of the absolute value of the Fourier transform of the process. If we denote the spectral density of white noise $|W(\mathrm{i}\,\omega)|^2 = q^2$, the spectral density of the process can be decomposed as (see [24] for an overview on the concepts in control theory)

$$S(\omega) = q^2 \, |G(\mathrm{i}\,\omega)|^2 = G(\mathrm{i}\,\omega) \, q^2 \, G(-\mathrm{i}\,\omega), \qquad (6)$$

where $G(\cdot)$ is typically referred to as the *transfer function* in signal processing. A transfer function corresponds to a stable system if and only if all of its poles (zeros of the denominator) are in the upper half of the complex plane. The (often non-trivial) procedure for finding a (stable) transfer function $G(\mathrm{i}\,\omega)$ is called spectral factorization.

An implication of Eqs. (5) and (6) is that, if we take the Fourier transform of $f \sim \mathcal{GP}(0, \kappa(r))$ and solve the Fourier transform of the process $F(\mathrm{i}\,\omega)$, we get $F(\mathrm{i}\,\omega) = G(\mathrm{i}\,\omega)\,W(\mathrm{i}\,\omega)$, where $W(\mathrm{i}\,\omega)$ is the (formal) Fourier transform of the white noise. This equation can be interpreted such that the process $F(\mathrm{i}\,\omega)$ is obtained by feeding white noise through a system with the transfer function $G(\mathrm{i}\,\omega)$ (see [56] for discussion on relation to temporal GPs). This takes an interesting role under the analysis of random networks in the sense that this relation is exactly what we want to capture in terms of the finding $\sigma(\cdot)$ w.r.t. Eq. (2) (feeding white-noise through the system formally corresponds to a network with Gaussian weights). Thus we can reduce the problem to analysing the transfer function $G(\mathrm{i}\,\omega)$.

The rest of this section is concerned with the particular case of the stationary Matérn family [61]:

$$\kappa_{\mathrm{Mat.}}(\mathbf{x}, \mathbf{x}') = \frac{2^{1-\nu}}{\Gamma(\nu)} \left( \sqrt{2\nu} \, \frac{\|\mathbf{x} - \mathbf{x}'\|}{\ell} \right)^{\nu} \mathrm{K}_\nu \left( \sqrt{2\nu} \, \frac{\|\mathbf{x} - \mathbf{x}'\|}{\ell} \right), \qquad (7)$$

where $\nu$ is a smoothness and $\ell$ a characteristic length-scale parameter, $\mathrm{K}_\nu(\cdot)$ the modified Bessel function, and $\Gamma(\cdot)$ the gamma function. For the Matérn class, the processes $f(\mathbf{x})$ are $\lceil \nu \rceil - 1$ times mean-square differentiable. In 1D, under the *angular frequency* convention (by Eq. (5)), we realize the spectral density function is of the following factorizable form (derivation in App. A.1):

$$S(\omega) \propto \left( \lambda^2 + \omega^2 \right)^{-(\nu+1/2)} = (\lambda + \mathrm{i}\,\omega)^{-(\nu+1/2)} \, (\lambda - \mathrm{i}\,\omega)^{-(\nu+1/2)}, \qquad (8)$$

where $\lambda = \sqrt{2\nu}/\ell$. Following Eq. (6), we can collect the transfer function of the corresponding stable part as $G(\mathrm{i}\,\omega) = (\lambda + \mathrm{i}\,\omega)^{-(\nu+1/2)}$. The remaining (power) spectral density (formally that of the driving white noise process $w(t)$ that captures the scaling coefficients) is $q^2 = 2\pi^{1/2}\lambda^{2\nu}\Gamma(\nu + 1/2)/\Gamma(\nu)$. Now taking the inverse Laplace transform of the transfer function $G(\mathrm{i}\,\omega)$ gives (this can be shown by considering the simplified expression $\mathcal{L}[x^{\alpha-1}\exp(-a\,x)](s) =$

$\Gamma(\alpha)\,(s+a)^{-\alpha}$, for $\alpha > 0$, details in App. A.2) and using the standard property of expanding the transform to the real line, gives us the transfer (or *activation*) in the input space:

$$\sigma(x) = \frac{q}{\Gamma(\nu+1/2)}\,\Theta(x)\,x^{\nu-1/2}\,\exp(-\lambda\,x), \tag{9}$$

where $\Theta(\cdot)$ is the Heaviside step function. For $\nu > 1$, these non-linear functions are smooth, continuous, and continuously differentiable. For $\nu < 1$, the functions are not smooth (rather taking the interesting form of an exponentially decaying step function), agreeing with the properties of the Matérn family. The function is not monotonic, which will be discussed in later sections. Fig. 3 shows realizations of $\sigma(\cdot)$ for various degrees of smoothness $\nu$. In terms of activation, the scaling can be made arbitrary, but scaling coefficients as in Eq. (9) can help with stabilizing training. With $\nu \to \infty$ Eq. (9) approaches a bell curve (see App. A.3 for a proof), recovering the relation between the Matérn family and the RBF kernel—giving the result of [68] (included in Figs. 3a and 3b).

## 3.2 Local Stationarity from Modulating Envelopes

Considering the activation function we have in Eq. (9) under the random network formalism of Eq. (2), we observe that this covariance function cannot be stationary due to the distributions of the weights being centered around zero, and hence no translational symmetry is present. Instead, the covariance is locally stationary (see discussion in [23]) in the same sense as the RBF-NN covariance function (see [53, 69]). This means that for any finite value of $\sigma_{\mathrm{b}}^2$ in $p(b) = \mathrm{N}(b\,|\,0,\sigma_{\mathrm{b}}^2)$ and $p(\mathbf{w})$ being binary white, we can write a composite covariance function with a Gaussian decay envelope

$$\kappa_{\text{Mat-NN}}(\mathbf{x},\mathbf{x}') \propto \exp(-\mathbf{x}^\mathsf{T}\mathbf{x}/2\sigma_{\mathrm{m}}^2)\,\kappa_{\text{Mat.}}(\mathbf{x},\mathbf{x}')\exp(-\mathbf{x}'^\mathsf{T}\mathbf{x}'/2\sigma_{\mathrm{m}}^2), \tag{10}$$

where $\sigma_{\mathrm{m}}^2 = 2\sigma_{\mathrm{b}}^2 + \ell^2$. As an example (and sanity check), Fig. 3b shows that we recover the exact covariance function $\kappa_{\text{Mat-NN}}(\cdot,\cdot)$ (dashed, barely visible due to almost exact match) using our proposed $\sigma(x)$ by MC integration in Eq. (3) (we use $K = 10{,}000$). For completeness, we also show the curves resulting from unit Gaussian weights on $p(\mathbf{w})$ (light colours).

## 3.3 Alternative View Through Green's Function

In the case of the isotropic RBF kernel, the network output is a linear combination of radial basis functions, as observed by Broomhead and Lowe [7] (see [66, 69] for discussion on the role of radial basis functions). This functional analysis view is taken by Poggio and Girosi [50] in deriving RBF networks. A similar construction could be applied here, but for the Matérn family this becomes unpractical, as explained below.

For any covariance function $\kappa(\mathbf{x},\mathbf{x}')$, we can define the associated covariance operator $\mathcal{K}$ as follows: $\mathcal{K}\phi = \int \kappa(\cdot,\mathbf{x}')\,\phi(\mathbf{x}')\,\mathrm{d}\mathbf{x}'$. For stationary covariance functions, this can also be written as a convolution. By assuming the inputs $\mathbf{x}$ to obey some density (see [70]), we consider an inner product defined by that density. Let the inner product be defined as $\langle f,g\rangle = \int f(\mathbf{x})\,g(\mathbf{x})\,w(\mathbf{x})\,\mathrm{d}\mathbf{x}$, where $w(\mathbf{x})$ is some positive weight function such that $\int w(\mathbf{x})\,\mathrm{d}\mathbf{x} < \infty$. In terms of this inner product, we define the operator $\mathcal{K}f = \int \kappa(\cdot,\mathbf{x})\,f(\mathbf{x})\,w(\mathbf{x})\,\mathrm{d}\mathbf{x}$. This operator is self-adjoint with respect to the inner product, $\langle \mathcal{K}f,g\rangle = \langle f,\mathcal{K}g\rangle$, and according to the spectral theorem there exists an orthonormal (in sense of $\int \varphi_i(\mathbf{x})\,\varphi_j(\mathbf{x})\,w(\mathbf{x})\,\mathrm{d}\mathbf{x} = \delta_{ij}$) set of basis functions and positive constants, $\{(\varphi_j(\mathbf{x}),\gamma_j)\}$, that satisfies the eigenvalue equation $(\mathcal{K}\varphi_j)(\mathbf{x}) = \gamma_j\,\varphi_j(\mathbf{x})$. Thus $\kappa(\mathbf{x},\mathbf{x}')$ has the series expansion (*cf.* Hilbert–Schmidt theorem) $\kappa(\mathbf{x},\mathbf{x}') = \sum_{j=1}^{\infty}\gamma_j\,\varphi_j(\mathbf{x})\,\varphi_j(\mathbf{x}')$.

In the case of the RBF kernel and assuming a Gaussian input density $w(\mathbf{x})$, these eigenvalues and eigenfunctions are available in closed form (see App. A.5). This directly relates to the envelope in the RBF-NN kernel, which takes the form $w^{1/2}(\mathbf{x})$ (see, [69], for a brief discussion). The relation to $\sigma(\cdot)$ is best understood through the so called Green's function $G(\cdot,\cdot)$ of the operator (the symbol re-used here on purpose), which is here defined through the property: $\mathcal{K}\,G(\mathbf{x},\mathbf{x}') = \delta(\mathbf{x}-\mathbf{x}')$ (for stationary systems, $G(\mathbf{x},\mathbf{x}') \triangleq G(\mathbf{x}-\mathbf{x}')$). From the definition it is apparent that $G(\cdot)$ generalizes the concept of impulse response in system theory to linear operators. Recalling that the transfer function is the Laplace transform of the impulse response, we see the relation between $\sigma(\cdot)$, Green's function, and Sec. 3.1. Solving the Green's function associated with $\mathcal{K}$ (and thus $\kappa(\cdot,\cdot)$) is non-trivial, and perhaps best grasped through the relation $G(\mathbf{x},\mathbf{x}') = \sum_{j=1}^{\infty}\gamma_j^{-1}\varphi^*(\mathbf{x})\,\varphi(\mathbf{x}')$ (see, *e.g.*, [66]). This relation helps theoretical understanding, but for deriving the activation function for, *e.g.*, the Matérn kernel

Table 1: Examples of UCI classification tasks, showing the Matérn-3/2 NN directly gives competitive accuracy, mean negative log predictive density (NLPD), and area under receiver operating characteristic curve (AUC) to sparse GPs or NN+GP hybrids. More results in App. B.2.

| (10-fold cv) | $n$ | $d$ | $c$ | NLPD | | | | ACC | | | |
|---|---|---|---|---|---|---|---|---|---|---|---|
| | | | | SVGP | GPDNN | SV-DKL | Matérn act. | SVGP | GPDNN | SV-DKL | Matérn act. |
| Adult | 45222 | 14 | 2 | .344±.006 | .435±.014 | **.316±.006** | **.316±.007** | .842±.005 | .821±.037 | **.855±.004** | .854±.005 |
| Connect-4 | 67556 | 42 | 3 | .629±.010 | .763±.018 | .459±.016 | **.450±.008** | .750±.006 | .768±.006 | .827±.009 | **.828±.004** |
| Covtype | 581912 | 54 | 7 | .494±.002 | .722±.025 | **.101±.005** | .118±.003 | .787±.002 | .842±.008 | **.962±.001** | .958±.001 |
| Diabetes | 768 | 8 | 2 | .506±.034 | .634±.012 | .691±.005 | **.486±.081** | .759±.056 | .744±.040 | .507±.143 | **.766±.044** |

| | $n$ | $d$ | $c$ | AUC | | | |
|---|---|---|---|---|---|---|---|
| | | | | SVGP | GPDNN | SV-DKL | Matérn act. |
| Adult | 45222 | 14 | 2 | .893±.004 | .774±.052 | .912±.003 | **.913±.004** |
| Connect-4 | 67556 | 42 | 3 | .824±.005 | .675±.019 | .909±.013 | **.913±.004** |
| Covtype | 581912 | 54 | 7 | .971±.001 | .943±.015 | **.998±.000** | **.998±.000** |
| Diabetes | 768 | 8 | 2 | .817±.049 | .769±.053 | .512±.095 | **.838±.051** |

this is highly impractical. In theory, one could resort to numerical approximations, but recovering $G(\cdot, \cdot)$ or $\sigma(\cdot)$ this way becomes unstable due to the inversion of the eigenvalues.

# 4 Experiments

We have included a comprehensive set of experiments that concentrate on analysing the quality of model prediction uncertainty and OOD tests. We consider illustrative toy data sets and standard classification benchmarks, analyse image classification, and finally provide a realistic safety-critical application example in radar emitter classification. The experiments were implemented in GPflow [43] (GPs and GPDNN), GPyTorch [22] (SV-DKL), and the rest in PyTorch (see App. B). For all neural network models using the Matérn activation functions the length-scale parameter $\ell$ is fixed as the preceding layer(s) take care of scaling the inputs, which serves the same purpose. App. B lists full details of all the experiments.

**Illustrative toy examples** In Fig. 1, we consider the binary Banana classification tasks under the presence of various GP priors. The inference is performed both with a variational GP (VGP, [43]) and using a small-size NN (to show noisiness) with one hidden layer (50 hidden units and activations corresponding to the top-row GP priors). Low predictive variance outside the training data is present in all methods, but the RBF and Matérn activations show localization around the training data. Fig. 2 shows the typical Bayesian deep learning toy regression task for in-between uncertainty prediction. Our activations capture similar behaviour as their GP counterparts, with 'stiffness' reducing with $\nu$.

**Benchmark classification tasks** In Table 1, we consider UCI benchmark classification tasks (including one small-data example) where we compare classification accuracy and negative log predictive density (NLPD) that penalizes both misclassification and miscalibrated uncertainty. We compare our model to NN+GP hybrids GPDNN [6] and SV-DKL [71], and a sparse GP method (SVGP, [33]). We use a Matérn $\nu = 3/2$ covariance function for all GPs and the corresponding activation. The NN architectures in all methods are the same (a fully connected network with layers d-1000-1000-500-50-c). We only consider the Matérn activation here, as, *e.g.*, the ReLU consistently performs poorly in terms of NLPD. The Matérn activation results are consistent and agree or outperform the hybrid methods that need to balance between training the NN and GP (which can be unstable and require approximate GP method in practice), when essentially having the same prior as our method. Details and further results for other activation functions can be found in App. B.2. We also include results for the SIREN [58] activation function, which induces a non-local infinitely smooth prior, comparable to the RBF.

**Out-of-distribution characterization with CIFAR-10** As a rule of thumb the uncertainty of OOD samples should be high and uncertainty of in-distribution samples should be low. This corresponds to predicting confidently only for samples within the domain familiar to the model through training. Fig. 4 shows the results for the experiments on the CIFAR-10 data set. Each model was trained with only images of five classes {plane, car, bird, cat, deer}. During testing, images from all 10 classes were present (now including also {ship, truck, frog, dog, horse}). In the histograms in Fig. 4, samples from classes present during training are marked with green and samples from classes not used in training are marked with red. The histograms use the standard deviation of the DNN output across MC dropout samples as the measure of uncertainty. Fig. 4 also shows examples of classified images.

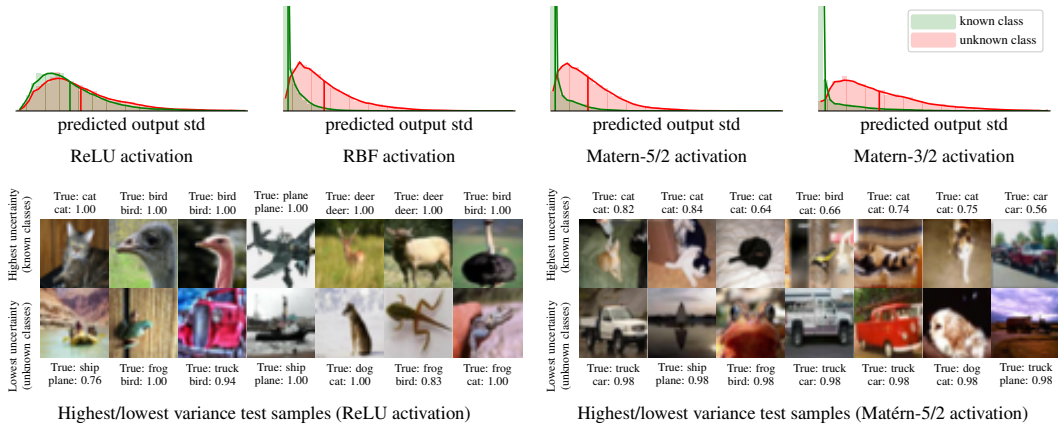

Figure 4: OOD example on CIFAR-10 with 5 classes ('known') in training and all 10 in testing. Top: Predictive variance histograms for known/unknown test class inputs, where the ReLU activation shows no separation, and the Matérn shows gradually more separation with decreasing $\nu$. Bottom: Test samples from ends of the histograms (true and predicted label + class prob.). For the Matérn, uncertain known and certain unknown samples feel intuitive (good calib.), while the results seem arbitrary for the ReLU. See App. B.3 for results for more activation functions.

The top row shows images from known classes with the highest uncertainty for both models and the bottom row shows images from unknown classes with the lowest uncertainty. For each image, the correct label is shown, along with the label predicted by the DNN and its probability. Both models use the GoogLeNet [64] CNN architecture with one additional fully connected layer before the final classification layer. The difference between the tested models is only the activation function used in this additional layer. For both models, pre-trained weights are used except for the additional layer and the final classification layer. Both have an accuracy of 97%. The Matérn-5/2 model gives a mean NLPD of $0.11$ on the test set known classes ($0.18$ for the ReLU).

**Black-box radio emitter classification** Radar emitter classification is a method for aircraft recognition in aviation. Classical methods for this problem resort to table lookup. The recent popularity and success of deep learning models have made them an appealing method to be applied to emitter classification. However, the lack of reliable uncertainty estimates for deep learning models risks making overconfident incorrect decisions with irreversible consequences. We train a CNN to classify simulated radar signals with a realistic radar library size training set with 100 different emitters. In the testing phase, the models receive samples from the same classes that were used in training and from additional unknown classes. These unknown classes are divided into two groups: classes that are *similar* to (and easily confused with) the classes used during training, and classes that are very distinct from the training data. Fig. 5 shows histograms of predictive entropy as the uncertainty measure for these groups of test data classes. The tested models differ only by the activation function used in the fully connected layer before the final classification layer, and have similar overall accuracy (59.3% vs. 59.5%). The Matérn activation functions, however, help the model have more appropriate uncertainty estimates.

## 5 Discussion and Conclusions

We established a link between neural network activation functions and the widely-known Matérn family of kernels (covariance functions). Our derivation took a control theory perspective to random neural networks, which has roots in early AI, but has been overlooked in recent years. Our experiments showed wide applicability and good practical performance in Bayesian deep learning tasks.

In the experiments, inference was done by MC dropout. We point out that even if recent studies have shown MC dropout to have poor uncertainty estimation on OOD inputs [55, 57, 59] or even in-distribution inputs [20], they used ReLU activation functions, where this behaviour is encoded into the model. In addition to MC dropout, Matérn activations can be used in conjunction with more advanced inference methods (*e.g.*, [41]). The selection of activation functions [52] is typically on the basis of accuracy alone, while we suggest that the choice should also be considered from an uncertainty quantification angle. However, the Matérn activation function is not monotonic, and thus

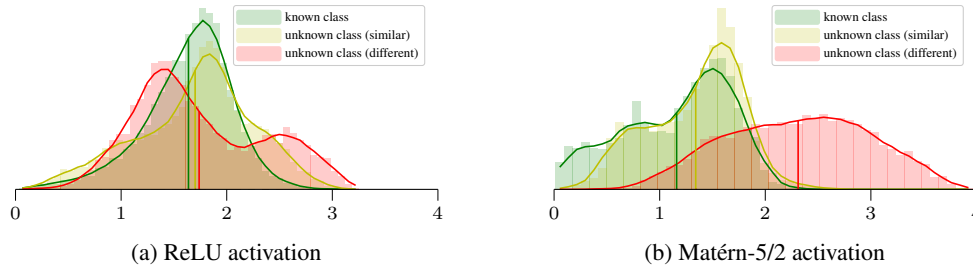

|  | (a) ReLU activation | (b) Matérn-5/2 activation |

Figure 5: Predictive entropy histograms for OOD radar emitter classification test samples, where the Matérn shows better separation both for the 'similar' and 'different' emitter sets.

the error surface associated with a single-layer model is not guaranteed to be convex [72]. It also saturates to $0$ for most $x$, and could thus be difficult to optimize [26]. Even so, usually choosing an appropriate learning rate was sufficient to achieve successfull training, and the only issues we encountered in the experiments occurred when using the (non-differentiable) exponential activation function. Both these properties are shared with RBF activations. We can, however, avoid some pathologies in RBFs. Stein [61] argues that an infinite smoothness assumption is unrealistic for modelling many physical processes, and Duvenaud [16] discusses several pathologies in infinitely smooth RBF and RQ models and points out that model misspecification is typically hard to spot. We thus consider the Matérn activations to be a convenient and principled building block for encoding continuity, (non-infinite) smoothness, and stationarity assumptions in neural networks.

Example codes implementing the proposed methods in this paper are available at `https://github.com/AaltoML/stationary-activations`.

## Broader Impact

We propose a new building block to help quantify uncertainty in deep learning. This contributes to creating methods in artificial intelligence that know what they do not know, which is important in safety-critical applications and in creating more robust and reliable systems. Such safety-critical applications include, for example, automatic medical diagnosis, self-driving vehicles, and general tasks for decision-making under uncertainty [2].

The contribution of this paper is in showing an explicit connection between two different paradigms in machine learning: we derive a non-linear activation function for neural networks which behaves as the widely used Matérn class of Gaussian process priors. This link can help build neural network models that are more robust and less vulnerable to out-of-distribution (OOD) data—which often occurs naturally in real-world settings [9, 54] or by potentially malicious construction, *e.g.*, adversarial attacks [63]—by making the neural networks behave more similarly to Gaussian process models, which have appealing properties in terms of well-calibrated uncertainty estimates and direct ways of including *a priori* knowledge, but typically do not directly scale to all kinds of applications and large data sets. However, we only show theoretical guarantees of this link to hold in the limit of infinitely wide neural networks with one hidden layer. In finite-size models we can only demonstrate the benefits empirically.

This paper is concerned with foundational research which is expected to have an impact across application areas. The example applications in the paper underline the wide variety of possible applications ranging from simple classification tasks to out-of-distribution class detection in image and radar emitter classification.

## Acknowledgments and Disclosure of Funding

We acknowledge the computational resources provided by the Aalto Science-IT project. This research was supported by the Academy of Finland grants 324345 and 308640, and Saab Finland Oy. We thank William J. Wilkinson, Paul E. Chang, and the anonymous reviewers for feedback on the manuscript. Additionally, we thank Henrik Holter for helpful discussions.

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
