[Supplementary Material]

# Supplementary Material: Stationary Activations for Uncertainty Calibration in Deep Learning

This supplementary document is organized as follows. App. A includes further details and derivations for the Methods section in the main paper. App. B includes details on the experiments, baseline methods, data sets, and additional tables and result plots.

## A  Derivations

### A.1  Spectral Factorization of the Matérn Spectral Density

We consider the stationary (and isotropic) Matérn covariance function (the difference here compared to the main paper is just defining $\mathbf{r} = \mathbf{x} - \mathbf{x}' \in \mathbb{R}^d$, the parameterization is the same as in [53]):

$$\kappa_{\text{Mat.}}(\mathbf{r}) = \frac{2^{1-\nu}}{\Gamma(\nu)} \left( \sqrt{2\nu} \, \frac{\|\mathbf{r}\|}{\ell} \right)^\nu K_\nu \left( \sqrt{2\nu} \, \frac{\|\mathbf{r}\|}{\ell} \right), \tag{11}$$

where $\nu$ is a smoothness and $\ell$ a characteristic length-scale parameter, $K_\nu(\cdot)$ the modified Bessel function, and $\Gamma(\cdot)$ the gamma function. This covariance function can be equally presented as a spectral density function, as discussed in the main paper (Wiener–Khinchin theorem). We use the angular frequency Fourier transform convention which simplifies keeping track of the scaling terms. From the Fourier-duality, the spectral density function of Eq. (11) can be recovered by the Fourier transform:

$$S_{\text{Mat}}(\boldsymbol{\omega}) = \int \kappa_{\text{Mat}}(\mathbf{r}) \, \exp(-\mathrm{i}\,\boldsymbol{\omega}^\mathsf{T}\mathbf{r}) \, \mathrm{d}\mathbf{r}. \tag{12}$$

Solving the integral gives

$$S_{\text{Mat}}(\boldsymbol{\omega}) = \frac{2^d \pi^{d/2} \Gamma(\nu + d/2) \lambda^{2\nu}}{\Gamma(\nu)} \left( \lambda^2 + \|\boldsymbol{\omega}\|^2 \right)^{-(\nu + d/2)}, \tag{13}$$

where $\lambda = \sqrt{2\nu}/\ell$. We can collect the constants into $q^2 = 2^d \pi^{d/2} \Gamma(\nu + d/2) \lambda^{2\nu} / \Gamma(\nu)$. We are interested in the response of the system under white noise, and thus we relax to $d = 1$. This is a recurring task in signal processing and control theory (see, [24] for a brief but comprehensive overview on the topic). Following the rationale in the main paper, we look at the transfer function that can be defined through the spectral density as $S(\omega) = G(\omega) \, q^2 \, G(\omega)^*$, where $[\cdot]^*$ is the complex-conjugate. Starting from Eq. (13), we can now do the spectral factorization by manipulating the expression (recall that $\mathrm{i}^2 = -1$ and $a^2 - b^2 = (a - b)(a + b)$):

$$S(\omega) = q^2 \left( \lambda^2 + \omega^2 \right)^{-(\nu + 1/2)} \tag{14}$$

$$= q^2 \left( \lambda^2 - (\mathrm{i}\,\omega)^2 \right)^{-(\nu + 1/2)} \tag{15}$$

$$= (\lambda + \mathrm{i}\,\omega)^{-(\nu + 1/2)} \, q^2 \, (\lambda - \mathrm{i}\,\omega)^{-(\nu + 1/2)} \tag{16}$$

$$= G(\mathrm{i}\,\omega) \, q^2 \, G(-\mathrm{i}\,\omega), \tag{17}$$

which is the form we use in the main paper.

### A.2  Recovering the Matérn Activation

Following Eq. (17), we can collect the transfer function of the corresponding stable part (see discussion in the main paper) as

$$G(\mathrm{i}\,\omega) = (\lambda + \mathrm{i}\,\omega)^{-(\nu + 1/2)}. \tag{18}$$

The remaining (power) spectral density (formally that of the driving white noise process $w(t)$ that captures the scaling coefficients) is $q^2 = 2\pi^{1/2} \lambda^{2\nu} \Gamma(\nu + 1/2) / \Gamma(\nu)$. Now taking the inverse Laplace

transform of the transfer function of $G(\mathrm{i}\,\omega)$ yields (this can be shown by considering the simplified expression $\mathcal{L}[x^{\alpha-1}\,\exp(-a\,x)](s) = \Gamma(\alpha)\,(s+a)^{-\alpha}$, for $\alpha > 0$)

$$\mathcal{L}^{-1}[G(s)](x) = \frac{1}{\Gamma(\nu+1/2)}x^{\nu-1/2}\,\exp(-\lambda\,x), \tag{19}$$

and by using the properties of the Laplace transform for expanding it to the real line, we recover the transfer (or *activation*) in the input space:

$$\sigma(x) = \frac{q}{\Gamma(\nu+1/2)}\,\Theta(x)\,x^{\nu-1/2}\,\exp(-\lambda\,x), \tag{20}$$

where $\Theta(\cdot)$ is the Heaviside step function.

### A.3    Recovering RBF Activations in the Limit of $\nu \to \infty$

From the empirical results in Fig. 3 it is clear that the Matérn activations of form Eq. (20) approach RBF activations as $\nu \to \infty$. However, it is not entirely trivial from Eq. (20) why this is the case. Thus we provide the following high-level proof.

Let $x \in \mathbb{R}_+$ and $\ell = 1$, such that Eq. (20) can be simplified to

$$\sigma(x) \propto x^{\nu-1/2}\,\exp(-\lambda\,x), \tag{21}$$

where $\lambda = \sqrt{2\nu}$. By rewriting

$$\sigma(x) \propto x^{\nu-1+1/2}\,\exp(-\sqrt{2\nu}\,x) \tag{22}$$

we can directly collect the terms in the following form ($\alpha = \nu + 1/2$ and $\beta = \sqrt{2\nu}$)

$$\sigma(x) \propto x^{\alpha-1}\,\exp(-\beta\,x), \tag{23}$$

which we recognize to have the same form as the probability density function of the gamma distribution

$$f(x\,|\,\alpha,\beta) = \frac{\beta^\alpha\,x^{\alpha-1}\,\exp(-\beta\,x)}{\Gamma(\alpha)}, \quad \text{for } x > 0, \alpha, \beta > 0, \tag{24}$$

where the parameterization is in terms of shape ($\alpha$) and rate ($\beta$). Because $\alpha \gg \beta$ ($\nu + 1/2 \gg \sqrt{2\nu}$) as $\alpha \to \infty$, we can leverage the known result that the gamma distribution tends to a Gaussian distribution as $\alpha \to \infty$ with mean $\alpha/\beta$ and variance $\alpha/\beta^2 \to 1/2$ and thus recovering the RBF activation in the limit

$$\lim_{\nu\to\infty} \sigma(x) = C\,\exp(-(x-c)^2), \tag{25}$$

where $c = (\nu+1/2)/\sqrt{2\nu}$ and $C$ is a positive constant. Formally this means that the RBF activation is pushed towards positive infinity (see Fig. 3a).

### A.4    Remark on the Role of $p(\mathbf{w})$

In the derivation for the RBF-NN kernel Williams [68] considers the prior $p(\mathbf{w})$ on the weights to be a delta distribution such that $\mathbf{w} = 1$. This is a special case of considering $\mathbf{w}$ to uniformly randomly get values in $\{-1, 1\}$ ('binary white'), which coincides with the presentation in Williams [68] and Rasmussen and Williams [53], due to the RBF activation function being an even function.

The common assumption of assuming $p(\mathbf{w})$ to be Gaussian, is covered in Fig. 3b (light coloured lines), where the smoothness properties (around origin for each $\nu$) are well preserved, but the tail behaviour is different (also agreeing with that of the RBF activation).

### A.5    Closed-form Expressions for the RBF Kernel

In Sec. 3.3, we went through an alternative view through functional analysis. This approach proved to be tricky for the Matérn class, even if it has been used for showing properties of the RBF (squared-exponential) kernel in the past. For the RBF many derivations simplify and can be done in closed form. This also helps in sanity checks for the Matérn class by comparing the limit behaviour to the

|  (a) Matérn-1/2  |  (b) Matérn-3/2  |  (c) Matérn-5/2  |

Figure 6: Gram matrices (colour map: ▨) corresponding to different values for $\nu$ in the Matérn-NN covariance function Eq. (10) with particular choices for the hyperparameters ($\ell = 0.5$ and $\sigma_\mathrm{b}^2 = 1^2$). The effect of the decay envelope is clearly visible when moving along the diagonal.

RBF. Thus we provide the following set of identities, which can be tedious to work out, but can be useful in analysing these problems in the spirit of Sec. 3.3.

For the RBF covariance function we can form the associated eigenbasis in closed form. We consider the re-parameterized RBF/squared-exponential kernel/covariance function of the form $\kappa(x, x') = \exp(-\alpha^2 |x - x'|^2)$, where $\alpha^2 = 1/2\ell^2$. And a weight function (input density function) $w(x) = \frac{\beta}{\sqrt{\pi}} \exp(-\beta^2 x^2)$. For these choices the eigenbasis of the covariance function (or the covariance operator) can be written as follows [18]:

$$\gamma_j = \beta \alpha^{2j} \left( \frac{\beta^2}{2} \left( 1 + \sqrt{1 + \left( \frac{2\alpha}{\beta} \right)^2} \right) + \alpha^2 \right)^{-(j+\frac{1}{2})}, \tag{26}$$

$$\varphi_j(x) = \frac{\sqrt[8]{1 + (\frac{2\alpha}{\beta})^2}}{\sqrt{2^j j!}} \exp\left( -\left( \sqrt{1 + (\frac{2\alpha}{\beta})^2} - 1 \right) \frac{\beta^2 x^2}{2} \right) \mathrm{H}_j\left( \sqrt[4]{1 + \left( \frac{2\alpha}{\beta} \right)^2} \beta x \right). \tag{27}$$

These eigenvalues and eigenfunctions are given in terms of physicists' Hermite polynomials $\mathrm{H}_j(\cdot)$. The obtained eigenfunctions are orthonormal with respect to

$$\int \varphi_i(\mathbf{x})\, \varphi_j(\mathbf{x})\, w(\mathbf{x})\, \mathrm{d}\mathbf{x} = \delta_{ij}. \tag{28}$$

For example, if we choose $\alpha = \sqrt{2}$ and $\beta = 1$, the first four eigenvalues and eigenfunctions are

$$\varphi_0(\mathbf{x}) = \sqrt[4]{3}\, e^{-x^2}, \qquad\qquad \varphi_1(\mathbf{x}) = \sqrt[4]{108}\, x\, e^{-x^2},$$

$$\varphi_2(\mathbf{x}) = \sqrt[4]{\frac{3}{4}}\, (6x^2 - 1)\, e^{-x^2}, \qquad\qquad \varphi_3(\mathbf{x}) = 3\sqrt[4]{3}\, (2x^3 - x)\, e^{-x^2}, \tag{29}$$

with associated eigenvalues $\gamma_0 = \frac{1}{2}, \gamma_1 = \frac{1}{4}, \gamma_2 = \frac{1}{8}$, and $\gamma_3 = \frac{1}{16}$.

Poggio and Girosi [50] do the derivation through Green's function for radial basis function networks, which recovers the RBF as the corresponding activation function. The RBF basis functions have a natural role, which can be seen through *Mercer's theorem* [44]. The theorem states that any positive-definite kernel can be represented as the inner product between a fixed set of features, evaluated at $\mathbf{x}$ and $\mathbf{x}'$:

$$\kappa(\mathbf{x}, \mathbf{x}') = \mathbf{h}(\mathbf{x})^{\mathsf{T}} \mathbf{h}(\mathbf{x}'). \tag{30}$$

The RBF kernel on the real line has a representation in terms of infinitely many radial-basis functions of the form $h(x) \propto \exp(-\frac{1}{4\ell^2}(x - c_i)^2)$, but any particular feature representation of a kernel is not necessarily unique (see, [45] and [17] for a more detailed overview).

# B   Experiment Details

The following sections provide further details on the experiment setup and implementation. Example codes are provided in separate files (see the accompanying `README` file) and they will also be available online at a later stage.

## B.1   Illustrative Toy Examples

To give an overview of the existing models and demonstrate how our model works, we provided two simple toy examples: a 2D classification and a 1D regression example.

**Classification**   In the classification example, we used the *Banana* data set (commonly even seen as a benchmarking data set). The Banana data set consists of 400 training samples, which are plotted as blue and orange circles in Fig. 1. The training set has 183 samples from class 0 (blue color) and 217 samples from class 1 (orange colour).

We used a fully connected neural network architecture with one hidden layer of 50 nodes. This choice was partly to highlight the differences to the 'infinitely wide' GP. For the same reason, we did not consider ensembling (the results were allowed to be noisy). The activation function of the hidden layer was set to represent the respective GP model kernel. For the Matérn activation function we fixed the lengthscale $\ell = 0.5$, which corresponds to scaling the inputs. The neural networks were trained for 2000 epochs with a batch size of 400 containing all training samples. The Adam optimizer was used with an initial learning rate of $0.02$, which was decayed by a factor 10 at epochs 250, 500, and 1000. Dropout with a rate of $0.2$ on the hidden layer was used during training to prevent overfitting and during testing to obtain MC dropout samples. For model testing a test sample grid of 300 by 300 samples in the range $(-3.75, 3.75)$ in both dimensions was created. To evaluate uncertainty during testing 5000 MC dropout samples were sampled for each grid sample (large number provides smoother figures). For each grid sample the MC dropout samples were averaged after applying the softmax function on the network output, and the resulting class probabilities $p$ and $q$ for the two classes were interpreted as a Bernoulli distribution standard deviation $\sigma = \sqrt{pq}$. This standard deviation was used as the uncertainty measure for the 2D classification task, plotted in Fig. 1. Figure Fig. 7 shows further tests on the *Banana* data set, visualizing the effect of changing the number of MC dropout samples and the number of neurons in the hidden layer of the network.

**Regression**   For the 1D regression example, we generated a simple data set using the function $f(x) = \frac{1}{4}(x - \frac{1}{2})^3 + \frac{1}{2}x^2 - x + \frac{1}{5}$, which was chosen to generate differing function values at the two clusters and to have moderate curvature in the function at both clusters. The clusters were generated by sampling 100 values for $x$ at both clusters (200 samples in total) from Gaussian distributions with means of $-1$ and 1, and with a standard deviation of $0.07$. The function $f(x)$ was evaluated at these values, and Gaussian noise with a standard deviation of $0.02$ was added. The resulting training samples for 1D regression are plotted as red dots in Fig. 2.

For simplicity, we used the same network architecture as in the classification example. Also the same dropout rate was applied during both training and testing. On the 1D regression task the network was trained for 2000 epochs with a batch size of 200 containing all training samples in a single batch. Adam optimizer with the same initial learning rate and learning rate decay schedule was used, as for 2D classification. The lengthscale was set to $\ell = 1$ also for this regression example. To test the model an evenly spaced sample grid of 100 samples in the range $(-2.5, 2.5)$ was used. For each grid sample 1000 MC dropout samples were obtained to estimate uncertainty. The standard deviation of the network outputs accross MC dropout samples was used as the uncertainty measure (sigma in Fig. 2, plotted as light blue color). The training and evaluation process was repeated 20 times for each model and the average result is reported in Fig. 2.

**GP baselines**   For the GP classification and regression baseline results we used GPflow 2 (`https://github.com/GPflow/GPflow`). The regression problem was a vanilla GP regression task, and for the classification task, we used the Bernoulli likelihood with VGP inference (corresponding to a non-sparse variant of [33]). GPflow has the ArcCos kernel built-in, and we implemented the ERF-NN and RBF-NN kernels following the parametrization in Rasmussen and Williams [53]. We used Adam for optimizing the hyperparameters and variational parameters.

Table 2: Further examples of UCI classification tasks, showing results for different choices for the covariance function (or correspondingly activation function). The setup in each sub table is the same and only the GP prior differs. Interestingly, the smoothness assumption of the prior seems to play a clear role, and the low-order Matérns perform clearly better than the RBF.

(a) RBF: The neural network model outperforms the other methods except on the adult data set. However, overall performance using the RBF is worse for all models compared to when a lower-order Matérn is used.

| (10-fold cv) | | | | SVGP | | GPDNN | | SV-DKL | | RBF activation | |
|---|---|---|---|---|---|---|---|---|---|---|---|
| | $n$ | $d$ | $c$ | NLPD | ACC | NLPD | ACC | NLPD | ACC | NLPD | ACC |
| Adult | 45222 | 14 | 2 | .341±.007 | .840±.006 | .431±.012 | .850±.005 | **.317±.006** | .854±.005 | .327±.012 | .853±.004 |
| Connect-4 | 67556 | 42 | 3 | .611±.009 | .756±.005 | .782±.026 | .762±.008 | .501±.110 | .811±.049 | **.473±.010** | .817±.005 |
| Covtype | 581912 | 54 | 7 | .505±.004 | .782±.002 | .824±.063 | .816±.021 | .125±.073 | .952±.029 | **.116±.002** | .959±.001 |
| Diabetes | 768 | 8 | 2 | .509±.037 | .755±.057 | .615±.015 | .741±.041 | .695±.005 | .576±.086 | **.565±.052** | .718±.047 |

(b) Matérn-5/2: The difference in the results given by this rather high-order Matérn and the RBF are rather clear. Overall, the performance is similar as when a Matérn-3/2 is used.

| (10-fold cv) | | | | SVGP | | GPDNN | | SV-DKL | | Matérn activation | |
|---|---|---|---|---|---|---|---|---|---|---|---|
| | $n$ | $d$ | $c$ | NLPD | ACC | NLPD | ACC | NLPD | ACC | NLPD | ACC |
| Adult | 45222 | 14 | 2 | .342±.007 | .841±.005 | .873±.116 | .804±.050 | **.315±.006** | .854±.005 | .317±.007 | .855±.004 |
| Connect-4 | 67556 | 42 | 3 | .619±.010 | .754±.006 | 1.74±.070 | .758±.009 | .462±.014 | .826±.006 | **.453±.009** | .827±.005 |
| Covtype | 581912 | 54 | 7 | .496±.003 | .786±.002 | .779±.032 | .822±.016 | **.101±.004** | .962±.002 | .115±.002 | .960±.001 |
| Diabetes | 768 | 8 | 2 | .507±.035 | .763±.055 | .608±.024 | .755±.042 | .693±.004 | .629±.073 | **.489±.074** | .768±.051 |

(c) Matérn-3/2: Neural network model performs comparably or better compared to the GP or NN+GP hybrids.

| (10-fold cv) | | | | SVGP | | GPDNN | | SV-DKL | | Matérn activation | |
|---|---|---|---|---|---|---|---|---|---|---|---|
| | $n$ | $d$ | $c$ | NLPD | ACC | NLPD | ACC | NLPD | ACC | NLPD | ACC |
| Adult | 45222 | 14 | 2 | .344±.006 | .842±.005 | .435±.014 | .821±.037 | **.316±.006** | .855±.004 | **.316±.007** | .854±.005 |
| Connect-4 | 67556 | 42 | 3 | .629±.010 | .750±.006 | .763±.018 | .768±.006 | .459±.016 | .827±.009 | **.450±.008** | .828±.004 |
| Covtype | 581912 | 54 | 7 | .494±.002 | .787±.002 | .722±.025 | .842±.008 | **.101±.005** | .962±.001 | .118±.003 | .958±.001 |
| Diabetes | 768 | 8 | 2 | .506±.034 | .759±.056 | .634±.012 | .744±.040 | .691±.005 | .507±.143 | **.486±.081** | .766±.044 |

(d) Exponential: Very similar results to when other Matérn options are used, except for our method, which suffers from the non-differentiability of the exponential activation function during training.

| (10-fold cv) | | | | SVGP | | GPDNN | | SV-DKL | | Exponential activation | |
|---|---|---|---|---|---|---|---|---|---|---|---|
| | $n$ | $d$ | $c$ | NLPD | ACC | NLPD | ACC | NLPD | ACC | NLPD | ACC |
| Adult | 45222 | 14 | 2 | .349±.006 | .839±.005 | .428±.014 | .852±.006 | **.315±.005** | .855±.005 | .522±.084 | .783±.151 |
| Connect-4 | 67556 | 42 | 3 | .680±.009 | .727±.005 | .747±.019 | .773±.006 | **.459±.015** | .827±.006 | .944±.056 | .670±.021 |
| Covtype | 581912 | 54 | 7 | .494±.002 | .791±.002 | .689±.128 | .811±.149 | **.101±.004** | .962±.001 | 1.20±.009 | .491±.009 |
| Diabetes | 768 | 8 | 2 | .508±.030 | .762±.061 | **.611±.013** | .736±.048 | .691±.004 | .533±.137 | 1.05±.307 | .443±.126 |

Table 3: UCI classification tasks with the SIREN activation function [58] showing NLPD, accuracy and AUC metrics. The imposed prior model is different, so the results are not directly comparable with the other methods (Table 2a being the closest).

| (10-fold cv) | | | | SIREN activation | | |
|---|---|---|---|---|---|---|
| | $n$ | $d$ | $c$ | NLPD | ACC | AUC |
| Adult | 45222 | 14 | 2 | .314±.006 | .854±.005 | .912±.004 |
| Connect-4 | 67556 | 42 | 3 | .449±.008 | .825±.005 | .909±.003 |
| Covtype | 581912 | 54 | 7 | .119±.002 | .957±.001 | .998±.000 |
| Diabetes | 768 | 8 | 2 | .487±.066 | .771±.054 | .835±.054 |

## B.2 Benchmark Classification Tasks

In the main paper, we considered four UCI benchmark classification tasks, where the aim was to compare our model to sophisticated previously published alternatives which aim to combine the flexibility of neural networks with GPs—in practice by taking neural network outputs as inputs to a GP model and performing joint learning and inference. We refer to these models as NN+GP hybrids. The rationale behind this experiment is that if our activations really emulate the behaviour of a GP with the corresponding covariance functions, we should be able to replace the hybrid GP part with a single layer with the corresponding Matérn activation. In the experiments, we also used a standard GP classifier as baseline (SVGP).

For benchmark classification tasks we chose four UCI data sets: 'adult', 'connect-4', 'covtype', and 'diabetes'. The adult data set contains information of US citizens and the classification task is to predict whether the individual makes over or under $50,000 in a year. The connect-4 data set contains

Figure 7: Illustrative comparisons on the *Banana* classification data set. The subfigures show decision boundaries and marginal predictive variance (low ▬▬ high) for MLP neural network results (one hidden layer) with the network activation function being the Matérn-$\frac{5}{2}$ activation. The rows show results with different numbers of MC dropout samples and the colums show results for different number of neurons in the hidden layer.

positions in the game of connect-4 and the classification task is to predict whether player 1 will win or lose, or if the game will end in a draw. The covtype data set contains samples with cartographic variables describing 30 by 30 meter forest patches, and the classification task is to assign samples into one of seven possible forest cover type classes. The diabetes data set contains diagnostic data for patients and the classification task is to determine whether the patient has diabetes or not. The small diabetes data set ($n = 768$) was chosen on purpose to see possible problems with small data.

The symbols $n$, $d$, and $c$ in Table 1, Table 2, and Table 3 represent the number of samples after removing missing values, the number of features in each sample, and the number of classes, respectively. 10-fold cross-validation was used to train and test each method. As preprocessing for all data sets categorical features were one-hot encoded and continuous features were normalized using standard scaling resulting in mean of 0 and variance of 1. For each fold of each method, the model was trained for 20 epochs with a batch size of 500 samples. For every method using a neural network as part of the model architecture, a fully connected network with layers d-1000-1000-500-50-c was used. The Adam optimizer was used as the model optimizer for all models except for SV-DKL, for which stochastic gradient descent optimizer was used with a weight decay of $0.0001$ and momentum of $0.9$. Training and testing was performed partially on CPUs and partially on GPUs (NVIDIA V100 and Tesla P100). On GPU the entire 10-fold cross-validation process for Matérn activation functions takes roughly 5 seconds for the diabetes dataset, 1 minute for the adult and connect-4 datasets, and 13 minutes for the covtype dataset. Training the GP models is considerably slower.

**SVGP** (Stochastic Variational Gaussian Process) is a sparse GP method [33]. We used a GPflow 2 [43] implementation of SVGP to perform benchmark classification tasks. The learning rate was set to $0.01$. Bernoulli likelihood was used at the model outputs if the number of classes for the data set was two, otherwise a softmax likelihood was used. The number of inducing points was set to 500. This model acted as baseline and should not be directly compared to the NN+GP models (thus results separated by a vertical line in the table).

**GPDNN**   (Gaussian Process Deep Neural Networks, [6]) is a NN+GP hybrid model. We modified the GPflow reference implementation that was provided by Bradshaw (`https://gist.github.com/john-bradshaw/e6784db56f8ae2cf13bb51eec51e9057`). We used a RobustMax likelihood with $0.1$ epsilon on the model outputs. The number of inducing points and the number of active dimensions for the GP were set to 50. The learning rate was set to $0.001$. Training the GPDNN was surprisingly fragile and without an exactly tuned learning rate and the epsilon value, the training diverged and NLPD values were very bad. The results in the tables are the best we could get for this model.

**SV-DKL**   (Stochastic Variational Deep Kernel Learning, [71]) is also a NN+GP hybrid model. The original SV-DKL publication considers a simple variant with pre-training and an additive GP. However, the SV-DKL framework has been improved as part of GPyTorch [22] to support general GP priors and end-to-end training. We used the reference implementation that is provided in the GPyTorch documentation (`https://gpytorch.readthedocs.io/en/latest/examples/06_PyTorch_NN_Integration_DKL/Deep_Kernel_Learning_DenseNet_CIFAR_Tutorial.html`). This implementation uses an approximate GP with inducing points placed on a grid. For the connect-4 data set a learning rate of $0.15$ was used, and for the other data sets the learning rate was set to $0.1$. Tuning the learning rates helped in achieving better convergence, but had only minor effects on the final numbers. The learning rate was used as is for training the neural network parameters, and when training the GP model parameters the learning rate was scaled by a factor of $0.01$. At 10 and 15 epochs, the learning rate was decayed by a factor of 10. The number of inducing points and the number of active dimensions for the GP were set to 50. A softmax likelihood was used at the model outputs. During training the likelihood was sampled 8 times to obtain an approximate distribution and during evaluation likelihood was sampled 16 times. Training the SV-DKL was rather robust compared to the GPDNN.

**Matérn activation functions (ours)**   were implemented using PyTorch. Similarly as in the simple examples, we fixed the scaling of the activations by choosing the lengthscale $\ell = 0.5$. The learning rates were set to $0.00005$, $0.0002$, $0.0005$, and $0.0001$ for data sets adult, connect-4, covtype, and diabetes, respectively. Individual learning rates were set to achieve better convergence on each data set. At 10 and 15 epochs, the learning rate was decayed by a factor of 10. A softmax function was applied at the network output layer to obtain class probabilities. Dropout with a rate of $0.2$ was applied on the layer with 50 nodes during training to prevent overfitting and during testing to obtain MC dropout samples. For each test set sample 100 MC dropout samples were obtained to estimate uncertainty in the outputs. After applying the softmax function on the MC dropout sample outputs, these 100 samples were averaged to obtain single class probabilities for each test sample.

**SIREN activation functions**   [58] were implemented similar to the Matérn activation functions using the same hyperparameters. The only difference to the Matérn activation function implementation was the activation function used in the last hidden layer of the neural network.

**Additional results**   Table 2 shows results on the benchmark classification tasks when different GP covariance functions and their corresponding activation functions are used. The table showing results for Matérn-3/2 is identical to Table 1 and shown here again for easier comparison. Looking at the results in this table we can observe that the neural network model using the activation function matching the GP kernel performs better or comparably to the GP and NN+GP hybrid models on all data sets, except when exponential kernel/activation is used. This is caused by the non-differentiability of the exponential activation which makes learning difficult for the neural network model. The GP and NN+GP hybrid models do not suffer from this problem as they are using the kernel instead of the activation function. Interestingly, the overall performance when an RBF is used is generally worse for all models when compared to most Matérn alternatives, which can be expected as discussed in Sec. 5. Table 3 shows results on the benchmark classification tasks when the SIREN activation is used.

To summarize, the results show that the Matérn activation is capable of capturing the behaviour, accuracy, and uncertainty quantification of the NN+GP hybrid models, but without the slightly awkward stack of combining models. In the NN+GP hybrids, approximate inference and possible sparse methods have to be applied anyway.

## B.3 Out-of-distribution Characterization with CIFAR-10

For image classification on the CIFAR-10 data set, we used the GoogLeNet [64] CNN architecture. We modified the standard architecture slightly by adding an additional linear layer of 512 nodes before the final classification layer. This makes the number of features in the end of the network be 1024-512-$c$, instead of 1024-$c$, where $c$ is the number of classes. We used pre-trained weights in every layer except in the additional linear layer and the final cltassification layer. Both the PyTorch base network architecture implementation and the pre-trained weights were based on those of `https://github.com/huyvnphan/PyTorch_CIFAR10`.

During training all pre-trained weights remained fixed. The difference between models compared on the CIFAR-10 task was only the choice of activation function in the additional linear layer of 512 nodes. Each model was trained for 100 epochs on the standard CIFAR-10 training set of 50,000 images but including only samples from five classes {plane, car, bird, cat, deer} resulting in 25,000 training samples being used. A batch size of 128 and a default learning rate of $0.01$ with the Adam (with weight decay) optimizer were used for training. For all Matérn activation functions a lengthscale $\ell = 1$ was used. During testing, images from all 10 classes of the standard test set of 10,000 images were present (now including also {ship, truck, frog, dog, horse}). Dropout was applied on the additional linear layer with probability $0.2$ both to prevent overfitting during training and to implement MC dropout during testing. During testing 10 MC dropout samples were obtained for each test sample. Both training and testing were performed on GPUs (NVIDIA V100 and Tesla P100).

The test set of 10,000 images has 5000 images from 'known' classes that were used during training and 5000 samples from 'unknown' classes. To measure model performance we calculate classification accuracy including only samples from known classes, since classification accuracy on unknown classes is inevitably 0% for all models. To further measure performance in terms of uncertainty estimates on known classes, we also calculate NLPD on test samples in known classes. To evaluate classification of test samples in unknown classes (where the standard NLPD measure is not applicable), we use a modified NLPD score:

$$\text{For the 'unknown' classes:} \qquad \text{NLPD}_{\text{mod}} = -\frac{1}{N} \sum_{i=1}^{N} \log \left( \frac{c}{c-1} (1 - y_{\text{max},i}) \right), \qquad (31)$$

where $N$ is the number of samples, $c$ is the number of classes, and $y_{\text{max},i}$ is the highest predicted class probability for sample $i$. This is similar to standard NLPD, differing only in how the predictive density is defined for unknown classes. The standard predicted class probability of the correct class can not be used, as the model can not predict any probability to unknown classes not specified for the model. Zero loss for a sample with this measure corresponds to predicting the maximum uncertainty prediction with class probabilities of $\frac{1}{c}$ for all classes, which is the best possible case for samples from unknown classes. This loss has infinite value for a sample, for which the model predicts some known class with 100% confidence, despite the sample belonging to none of the known classes. This is not a standard metric, but provides interesting and intuitive numerical comparison for classifying samples from unknown classes.

The numerical results for using different activation functions in CIFAR-10 classification are shown in Table 4. For the results in the table, the network outputs were averaged across the 10 MC dropout samples and a softmax function was applied on the resulting mean network outputs to obtain class probabilities. Looking at the classification accuracy values, different models perform very similarly achieving very high accuracy except for when the exponential (Matérn-1/2) or step activation is used. The model with the exponential activation manages to learn something meaningful with an accuracy of 37.6%, but the step activation has practically the accuracy of a random guess across the 5 known classes. This bad performance is explained by the non-differentiability of the Matérn-1/2 (see Fig. 3a) and step activations, which makes learning very challenging for the neural network.

In terms of NLPD on the known classes, the Matérn class of activation functions have the best performance, but quite closely followed by the ERF and ReLU activations. For a practical example, a sample will have an NLPD of $0.1$ if it is predicted correctly with 90% class probability. The NLPD values on the unknown class samples are strongly in favour of the RBF, Matérn-5/2, and Matérn-3/2 activations. The step function has the lowest NLPD on unknown class samples but this is arbitrary as the model using the step function has not learned anything meaningful based on classification

Table 4: CIFAR10 classification accuracy (on the 'known' classes that were used during training) and mean NLPD values reported separately for the 'known' classes and 'unknown' OOD classes. For accuracy, larger is better, and NLPD smaller is better. For unknown classes the NLPD is calculated as described in App. B.3.

|  | RBF | Matérn-5/2 | Matérn-3/2 | Exponential | ERF | Step | ReLU | SIREN |
|---|---|---|---|---|---|---|---|---|
| Accuracy (known class) | 0.973 | 0.974 | 0.973 | 0.376 | **0.975** | 0.217 | 0.973 | 0.972 |
| NLPD (known class) | **0.099** | 0.110 | 0.103 | 1.26 | 0.147 | 1.52 | 0.181 | 0.106 |
| NLPD (unknown class) | 0.801 | 0.754 | 0.896 | 1.22 | 2.73 | **0.114** | ($\sim$inf) | 1.62 |

accuracy and is just predicting all classes with almost equal probabilities, as can be seen in the predicted probabilities for the step activation in Fig. 8.

After step activation, the second lowest NLPD value on unknown class samples is achieved by the Matérn-5/2 activation at $0.754$. This value corresponds to predicting unknown samples with a maximum class probability of around 62% on average. For comparison, the NLPD of $2.73$ for ERF corresponds to a maximum predicted class probability of 95% on average for unknown samples. For ReLU activation the NLPD for unknown samples is infinite within numerical limits as the model predicts unknown class samples with 100% confidence to belonging to some of the known classes.

The uncertainty histograms and examples of classified images in Fig. 8 show improvement in OOD sample uncertainty estimates when Matérn activations are used. For the results in this figure, the uncertainty measure used was the standard deviation of MC dropout samples at the classification layer outputs before applying the softmax function. Ideally, we would like to be able to separate known and unknown class samples from each other based on their associated uncertainty. The histograms in Fig. 8 show that Matérn class activation functions show good separation between samples of unknown and known classes. The ERF activation function also shows decent separation. The ReLU doesn't show practically any separation in uncertainty of the unknown and known classes. For the exponential and step activations, the histograms are not very meaningful as with these activations the models didn't manage to learn to properly classify the sample images. Looking at the samples of classified images for ReLU, all known class samples with highest uncertainty have a predicted class probability of 1.00 and many of the unknown class samples with lowest uncertainty have quite low predicted class probabilities. This suggests that for ReLU, standard deviation of MC outputs and the predicted class probability do not correlate very well with each other as uncertainty measures.

## B.4 Black-box Radio Emitter Classification

In the final example, we consider an application outside the standard benchmarking tasks, which underlines both the importance of uncertainty quantification and OOD characterization. The data set for the radio emitter classification task was generated through simulations, as large radar data sets are not publicly available. However, the properties are well understood and the observations can be simulated.

**Data** We use a simulator provided by Saab for creating a realistic data set. The simulation setup was tuned in collaboration with experts at Saab. Each simulated radar emitter sample has a carrier frequency, pulse width, and a series of 25 to 250 pulse arrival times. The pulse arrival times follow some pattern characteristic to the specific radar emitter. To set up a classification task we specify 100 simulated emitters with characteristic carrier frequencies, pulse widths, and pulse arrival time patterns. These characteristic parameters are partly overlapping between different simulated emitters to make the classification task harder. Each simulated emitter can also operate in 10 different modes of operation, each of which have slightly different characteristic parameters. For each mode of each emitter, we simulated 5 samples on 11 different noise levels. In total, this adds up to 55,000 training samples. The noise is added to the samples by disturbing the series of pulse arrival times, by either dropping pulses or by adding spurious pulses to the sequence.

In addition to the training set, we simulated a test set with three groups of samples. The first test sample group of 10,000 samples was generated from the same simulated emitters as the training set. The second test sample group of 10,000 samples was generated from simulated emitters that are unknown to the model, but that have their characteristic parameters resembling some of the simulated emitters in the training set, making these test samples hard to distinguish from familiar training samples. The third test sample group of 10,000 samples was generated from simulated emitters that

are unknown to the model, and that have very different characteristic parameters to the simulated emitters in the training set, making these test samples easy to distinguish from samples that are from the training set emitters. Together these three test sample groups help to set up an OOD classification test, for which an ideal model can separate the samples in the two groups generated from unknown emitters, from the samples generated by the familiar training emitters. In this task separating the unknown emitter samples that more closely resemble the training set emitter samples should be harder than separating the samples that are from emitters with very different characteristics.

**Model** The neural network architecture used for this task is a CNN architecture with skip connections, combined with a small parallel fully connected network. The CNN part of the network is used to process the series of pulse arrival times with 1D convolutional layers, and the fully connected part is used to process the carrier frequency and pulse width information. The outputs of these two parallel networks are concatenated and a fully connected layer structure of 148-120-$c$ is used to provide the final classification ($c = 100$). Dropout with a rate of $0.2$ is applied on the fully connected layer with 120 nodes. The difference between the two tested models is also the activation function used in this fully connected layer: either a ReLU activation or Matérn-5/2 activation with a lengthscale $\ell = 1$ was used. Dropout is active also during testing to provide MC dropout samples (100 MC dropout samples are obtained). The Adam optimizer with a learning rate of $0.01$ was used in training. The models are trained for 7 epochs with a batch size of 50 samples.

Highest uncertainty (known classes)

True: bird / bird: 0.86 | True: cat / cat: 0.73 | True: cat / cat: 0.81 | True: car / car: 0.94 | True: cat / cat: 0.60 | True: car / car: 0.70 | True: cat / cat: 0.71

Lowest uncertainty (unknown classes)

True: dog / cat: 0.98 | True: truck / car: 0.99 | True: dog / cat: 0.98 | True: dog / cat: 0.98 | True: dog / cat: 0.98 | True: truck / plane: 0.99 | True: frog / cat: 0.98

known class / unknown class

predicted output std

Highest/lowest variance test samples (RBF activation)

(a) Results with RBF activation

Highest uncertainty (known classes)

True: cat / cat: 0.82 | True: cat / cat: 0.84 | True: cat / cat: 0.64 | True: bird / cat: 0.66 | True: cat / cat: 0.74 | True: cat / cat: 0.75 | True: car / car: 0.56

Lowest uncertainty (unknown classes)

True: truck / car: 0.98 | True: ship / plane: 0.98 | True: frog / bird: 0.98 | True: truck / car: 0.98 | True: truck / car: 0.98 | True: dog / cat: 0.98 | True: truck / plane: 0.98

predicted output std

Highest/lowest variance test samples (Matérn-5/2 activation)

(b) Results with Matérn-5/2 activation

Highest uncertainty (known classes)

True: bird / bird: 0.84 | True: bird / bird: 0.49 | True: bird / bird: 0.55 | True: cat / cat: 0.63 | True: bird / bird: 0.45 | True: bird / deer: 0.52 | True: plane / car: 0.71

Lowest uncertainty (unknown classes)

True: truck / plane: 0.99 | True: truck / car: 0.99 | True: dog / cat: 0.99 | True: dog / cat: 0.99 | True: truck / car: 0.99 | True: horse / deer: 0.98 | True: dog / cat: 0.98

predicted output std

Highest/lowest variance test samples (Matérn-3/2 activation)

(c) Results with Matérn-3/2 activation

Highest uncertainty (known classes)

True: deer / deer: 0.79 | True: deer / deer: 0.65 | True: deer / deer: 0.62 | True: deer / deer: 0.68 | True: deer / deer: 0.86 | True: deer / deer: 0.76 | True: deer / deer: 0.73

Lowest uncertainty (unknown classes)

True: horse / car: 0.27 | True: horse / deer: 0.27 | True: dog / deer: 0.30 | True: horse / deer: 0.28 | True: ship / plane: 0.41 | True: horse / deer: 0.33 | True: truck / car: 0.69

predicted output std

Highest/lowest variance test samples (Exponential activation)

(d) Results with the exponential (Matérn-1/2) activation

(e) Results with ERF activation

(f) Results with step activation

(g) Results with ReLU activation

Figure 8: Additional OOD example on CIFAR-10 with 5 classes ('known') in training and all 10 in testing. Left: Predictive variance histograms for know/unknown test class inputs, where the Matérn shows gradually more separation with decreasing $\nu$, except for the exponential activation which shows poor separation. Right: Test samples from ends of the histograms (true and predicted label + class prob.). For the Matérn-3/2, Matérn-5/2, RBF, and ERF uncertain known and certain unknown samples feel intuitive (good calibration), while the results seem arbitrary for the exponential, ReLU and step activations.