[Reviews · NeurIPS 2020]

Review 1

Summary and Contributions: This paper introduces a method to derive activation functions from stationary Matern family kernels, allowing Bayesian deep nets to approximate uncertainties captured by a GP.

Strengths: The research topic looks interesting. To my knowledge, most recent efforts to connect NNs and GPs are spent to build links between kernels and NN extracted features (e.g. deep kernels, kernel feature expansions), it is hence nice to have another view on the activation functions.

Weaknesses: 1. While this paper describes a method to derive activation functions from kernels, it seems not covering much work on kernel feature expansion (e.g. random Fourier features). Given that the proposed approach is also based on the Fourier duality, it would be good to claim further differences between performing it at a feature level and performing it at an activation function level. 2. Although I appreciate this paper has included various tasks to show the goodness of approximation, some of the experiments can still be tuned to give better illustrations. For instance, a) to include methods based on kernel feature expansion as discussed above. b) both figure 1 and figure 2 shows the approximated distributions, it should be possible to perform quantitive comparisons on such toy datasets (e.g. compute divergence/discrepancy). It seems the approximation depends on the number of hidden units as well as the number of Monto-Carlo samples. c) For later out-of-distribution detection task, it would also be better if some results from the original GPs are included (maybe not on a huge image dataset considering computational costs).

Correctness: From a top-level check, the proposed method seems to work, while the experiments can be improved as discussed above. On the claim side, one issue is the "uncertainty calibration" claimed in the title and through the paper. Recently this term has a particular meaning in both classification and regression tasks regarding probability calibration. It will require a set of definitions and evaluation metrics to verify the level of calibration, which is not seen in this paper. I would be more comfortable if the author can rename it to "uncertainty quantification", as the calibration part is not really touched in this paper.

Clarity: Yes, readers with related background should get most contents without issues.

Relation to Prior Work: As discussed above, my only question is about the differences between approximate GPs using kernel features and using the proposed method.

Reproducibility: Yes

Additional Feedback: As shown by my score, I generally like the idea and method, it would be better if the authors can address my problems above regarding kernel extracted features and kernel extracted activation functions for GP approximation. ===========AFTER AUTHOR FEEDBACK===================== The author feedback provides some initial answers to my question. I still think it might require some further work to explain the differences experimentally before I can vote for an apparent accept. I would encourage the authors to includes a detailed discussion and experiments on related work regarding kernel approximations so we can see the pros and cons.


Review 2

Summary and Contributions: This paper presents a novel neural network activation function that resembles some properties of the Matern kernel popular in Gaussian processes (GPs).

Strengths: The paper is well written and easy to follow. The best thing about the paper is it variously discusses the connections of the proposed method to other techniques and disciplines which, unfortunately, is rare in many machine learning papers nowadays.

Weaknesses: As highlighted in the additional feedback sections below, * The motivation and impact are not clear. * The paper requires more experimental evaluations.

Correctness: Technically correct though empirical validations require further work.

Clarity: Well-written.

Relation to Prior Work: The provided relations are clear. I have pointed out many other in the additional feedback section below.

Reproducibility: Yes

Additional Feedback: Thanks for the supplementary materials! The overall motivation is not clear. It is true that Matern kernels are good at capturing sharp transitions, as shown in Fig 1. There are many other methods to achieve similar, if not better, results. For instance, we can learn kernels [1,2], use deep kernels [2], use spectral mixture kernels [3], use "neural-network kernels" [4], etc. Comparisons with [1]-[5] would provide further insights. Please also report MSE and AUC in addition to the accuracy. It is not clear why the paper discusses stationary kernels. Stationarity is indeed a limitation of classical kernel methods. Easy to use recent methods that capture nonstationarity provide better estimates (mean and variance), even in real-world examples [5]. It is not clear why the stationarity is highlighted as an advantageous property. It is not clear why MC dropout is used instead of, say, more stable SVI. It is true that MC dropout typically provides incorrect variance estimates for both in and out of distribution data and the accuracy is sensitive to the type of activation function. To the best of my knowledge, there is no clear explanation for why this happens. It is not clear why using the proposed activation function leads to better performance, especially for OOD. OOD performance has also not been adequately benchmarked as in, say, [6]. The metrics and the techniques provided in [6] can be used. Can the recently proposed activation functions such as SIREN [7] achieve similar results? Other than the robustness against OOD samples, how would the paper be more impactful and beneficial to the NeurIPS readership? [1] Black-box Quantiles for Kernel Learning, AISTATS’19 [2] Deep Kernel Learning, AISTATS’16 [3] Spectral Mixture Kernels for Multi-Output Gaussian Processes, NeurIPS’17 [4] Gaussian processes for Machine learning [5] Automorphing Kernels for Nonstationarity in Mapping Unstructured Environments, CoRL’18 [6] Simple and Scalable Predictive Uncertainty Estimation using Deep Ensembles, NeurIPS’17 [7] Implicit Neural Representations with Periodic Activation Functions POST-REBUTTAL COMMENTS I do appreciate the idea of introducing Matern kernels as a new activation function for NNs. However, in my opinion, the contributions of the paper, in its current form, are not sufficient enough to accept the paper. Since the majority of text is simply explaining existing work, I believe that experiments should be strong enough, benchmarking against different techniques and using different datasets. This is specifically important because the authors claim that the proposed activation is superior that existing NN activation functions. Note that this is a strong claim and would be revolutionary. This is the main reason why I hesitate accepting the paper straightaway and demand more comparisons on diverse datasets and existing methods that can achieve similar results. I believe providing authors a chance to improve their paper and submit a well matured version to a different venue will result in a better validated and more impactful paper. The readership of a conference such as AISTATS would appreciate this paper more. Answers to the rebuttal: 1. "Stationarity encodes conservative behaviour suitable for uncertainty quantification" - It is not clear what this means. As I have highlighted in my original review, stationary is actually a limitation than a feature. 2. "...tackling a different problem, where the kernel is not used for encoding specific prior information, but inferred from data..." - I do not fully agree with this statement. Since a kernel measures similarity, it always encode prior information. This is specifically obvious in geological mapping (e.g. kriging) and robotics applications (citations provided in the original review). Authors also agree in the rebuttal that "the choice of kernel/activation function is up to the modelling task and expert knowledge." Learning a generic kernel (See [1,3]) is always better than using a Matern kernel. 3. It is still not clear why MC dropout is used instead of, say, more stable variational inference. Also, the OOD performance has not been quantitatively evaluated as in [6].


Review 3

Summary and Contributions: In the paper, authors proposed a new activation method derived from the Matern kernel family. Besides the thorough analysis and explicit explanation of their method, a main contribution is the thought of leveraging the link between Gaussian process methods and neural network, which is inspiring to the community.

Strengths: The paper has a solid theoretical standing and explicit explanation.

Weaknesses: Motivations of using Matern family should be further clarified though I agree Matern family is the reasonable choice in a neural network. Since there are many GP kernels and some of them have similar theoretical consideration with the Matern family, a broader discussion may be added.

Correctness: Yes, the claims, method and the expirical methodology are correct.

Clarity: Yes, the idea, method and explanation are well presented.

Relation to Prior Work: Related work focuses on OOD detection.

Reproducibility: Yes

Additional Feedback: (1) I've seem several recent work focusing on the Bayesian network so that the learning process is calibrated with uncertainty estimation. Comparing with the work such as the "Deep Neural Networks as Gaussian Processes", what's the difference, benefit and limitations of only using a Matern activation function? (2) I need more motivation explanations on choosing the Matern kernel. Matern kernel is oriented for the purpose of a better depiction of the physical process(Stein's kriging work, or Cressie and Christ. ). We can take it as the solution of a SPDE. I'm not sure if authors have analyzed the effectiveness of a general SPDE form for activation function. I personality feel that will be a higher-level theory. There are several kernel functions having a better capability of capturing intrinsic features from the data, i would like to see some comparisons with some of them. (3) what's the complexity of using the proposed activation function? Any efficiency comparison results? (4) A naive question, why not Bayesian network if uncertainty estimation is considered important? or Deep kernel learning (wilson, zhiting 2015) ? I am always thinking a neural network calibrating uncertainty with kernel tricks is essentially a simplified Bayesian network. ######################################## I like the comparison with SIREN in the feedback, I would suggest the author to add it to the supplementary if it is possible. The motivation seems a common concern among reviewers, On the whole, I think it is a very good submission.


Review 4

Summary and Contributions: This paper introduces a new set of activation functions for DNN that are based on the Matérn family of kernels in Gaussian process. The paper argues that these new activation functions allow one to impose stationarity, continuity, and various degrees of differentiability as priors on the resulting function. Such priors are beneficial for calibrating out-of-distribution uncertainty.

Strengths: + The motivation for the proposed activation functions is well formulated + The theoretical grounding appears sound + The empirical results illustrate how the proposed approach provides better OOD uncertainty on multiple datasets

Weaknesses: - As noted by the authors, the inference procedure used in this work has not been given a substantial amount of attention (it offers a direction for future work) - I found some of the empirical results a little hard to interpret

Correctness: The claims and methods appear correct to me.

Clarity: The paper is very well written.

Relation to Prior Work: The relations to previous work have been extensively discussed (though I'm not an expert in the field and cannot tell if the paper is missing some relevant references)

Reproducibility: Yes

Additional Feedback: I really enjoyed reading this paper. The motivation is well presented and the choice of methodology seems well-justified. I like that, unlike in some previous work, this work is based on changes to the model rather than, for example, the optimisation procedure in order to achieve better OOD uncertainty. I found Fig.1 somewhat hard to interpret. As expected for GPs with a stationary kernel, the top row illustrates how away from the data the uncertainty increases quite fast. However, in the bottom row the uncertainty away from the data does not increase uniformly and there are still large areas for which the uncertainty appears similar to that in the parts of the domain with the data. Is it fair to say that achieving the same error paterns in both rows in this figure (and I'm mostly refering to the Matern 5/2 case here) would be the ultimate goal in your case? Looking at Fig. 2, it appears that in all cases of NNs (in the second row) the noise that's estimated in parts of the domain with the data is higher than in the GP cases, i.e. the NNs seem to estimate higher observation noise (epsilon, where y = f(x) + epsilon) than the GPs. Consequently (for this and possibly other reasons), the uncertainty in the parts of the domain with no data is also higher in the NN cases than for the corresponding GPs. What is the reason for the higher observational noise in the case of NNs? Also, for the Matern kernels in the NN cases, it appears that the predictions converge to the mean of the data far from the data (in this case, on the sides of the domain). However, that doesn't seem to be the case in the RBF case. Do you know why this is? Unless I missed it, you don't seem to discuss the role of the lengthscale l of the Matern kernel in any of the experiments. Looking at Fig. 2, it appears that the lengthscale might be very short for the NNs. Is there a prior on this parameter? Is it being learnt or fixed to some value? Could you also comment further on the role of this parameter in the larger examples? Given that the activation functions are not monotonic, one might expect the training to be a lot more prone to local minima and generally harder to train (which you allude to in the discussion). A more thorough discussion of this potential limitation would be interesting, though this could be phrased as a direction for future work. In general, I'd be happy to increase my score if you provided some intuition on the points I raised above. Thanks. Minor: Line 300 - the a

[Author Response · NeurIPS 2020]

**Author response: 'Stationary Activations for Uncertainty Calibration in Deep Learning'**      **NeurIPS: #5154**

We thank the anonymous reviewers for their enthusiasm and detailed comments on the manuscript. We summarise the
reviews as positive, and the main concerns were related to clarifying the motivation and the experiments. We agree with
the requests, and we will use the additional ninth page in the camera-ready paper for expanding the details as requested.
We start by addressing R2's concerns as they had the lowest score.

**R2: (1)** The main motivation for this work is to establish understanding about the link between Matérn GP priors and
neural network activation functions. This link is *explicit* as shown in this paper. A motive for this is to allow NN models
to incorporate some of the appealing properties of GP models (*e.g.*, well-characterized uncertainties), while maintaining
the flexibility and efficiency of NNs. The choice of kernel/activation function is up to the modelling task and 'expert
knowledge'. We merely provide a building block. The Matérn is a widely used prior, and worth adding to the NN tool set.
Stationarity encodes conservative behaviour suitable for uncertainty quantification (see R4(2)). **(2)** The references R2
provides ([1–3, 5]) are tackling a different problem, where the kernel is not used for encoding specific prior information,
but inferred from data (*cf.*, 'automatic statistician'), and [4] is covered in this paper (limiting RBF case and NN kernel).
**(3)** In the experiments, we originally reported only accuracy
and NLPD (accounting for uncertainty). As requested, we
added AUC to the results, which is in-line with the previous
results (see table), with our proposed model outperforming
the baselines. Comparison to SIREN is interesting (NB: the

UCI classification tasks with Matérn-3/2 activation/kernel showing the AUC metric. Also results for SIREN activations included.

| (10-fold cv) | | | | SVGP | GPDNN | SV-DKL | Matérn activ. | SIREN activation | | |
|---|---|---|---|---|---|---|---|---|---|---|
| | $n$ | $d$ | $c$ | AUC | AUC | AUC | AUC | NLPD | ACC | AUC |
| Adult | 45222 | 14 | 2 | .893±.004 | .774±.052 | .912±.003 | **.913±.004** | .314±.006 | .854±.005 | .912±.004 |
| Connect-4 | 67556 | 42 | 3 | .824±.005 | .675±.019 | .909±.013 | **.913±.004** | .449±.008 | .825±.005 | .909±.003 |
| Covtype | 581912 | 54 | 7 | .971±.001 | .943±.015 | **.998±.000** | **.998±.000** | .119±.002 | .957±.001 | **.998±.000** |
| Diabetes | 768 | 8 | 2 | .817±.049 | .769±.053 | .512±.095 | **.838±.051** | .487±.066 | .771±.054 | .835±.054 |

SIREN paper was put on arXiv *after* the submission DL), and to answer your question, we ran the experiments with it
as well (see table). On the UCI tasks, SIREN performs comparably to our method. On the OOD image classification
task, it performs clearly worse (known-class NLPD: 0.103 vs. 0.106, unknown-class NLPD: 0.896 vs. 1.62), but still
better than the baselines. Note that SIREN encodes a different type of prior (infinite smoothness, like the RBF).

**R1: (1)** We are glad that the question about relation to kernel feature expansions was brought up. Fourier features
(random, dense/structured, sparse) are typically leveraged for stationary kernels by projecting the GP problem on a set
of harmonic basis functions. While we share the idea of using the Fourier duality, the resulting model is spanned by
different basis functions; *e.g.*, sinusoidal FFs enforce (global) stationarity (approximation is based on Eq. (5)), while
our approach is *locally* stationary as defined by the Gaussian weights in Eq. (2), which the approximation is based on.
This discussion was left out in the interest of space, but the additional page will give us space to cover this. **(2)** We
appreciate the suggestions for improving the visualizations. We did focus on real data (with three different real-world
experiment setups) in our quantitative experiments, and the toy data examples were to give a general understanding.
Your suggestion of varying the number of hidden units and MC samples is good and easy to do. We'll include this in the
appendix to facilitate understanding of the effect of these parameters. We'll also run a GP on CIFAR-10 as suggested.
**(3)** We recognize your concern with the term 'uncertainty calibration' (used here as GPs are commonly said to have
representative uncertainties). We will replace it with the less loaded 'uncertainty quantification' where applicable.

**R3: (1)** The paper 'Deep Neural Networks as Gaussian Processes' (thank you for the reference) works in the scope
of our Sec. 2 (connection from activation functions to GP kernels for principled inference). We take the opposite
direction (GP prior $\rightarrow$ activation function) to encode properties of the GP prior into the NN. **(2)** Indeed, the SPDE link
for constructing Gauss–Markov random fields from stationary kernels is relevant here. The deeper-level connection is
related to the spectrum of the corresponding 'covariance operator' (the covariance function is formally the kernel of this
operator). We agree that there is still a lot to uncover in this space. **(3)** The computational complexity of our model is
on par with using other activation functions (say, a ReLU) in the NN models. In practice, we may converge slightly
slower (*e.g.*, 53 s vs. 42 s for the results on the 'adult' data set). Compared to the GP models, especially with large
data sets, we gain a considerable speed-up. **(4)** You are right that in this regard our approach can be thought of as a
simplified Bayesian neural network—but yet as one, where the Matérn prior can be directly encoded.

**R4: (1)** We understand your comment on not dedicating attention to the inference methods (also raised by **R2**). This
is true and mostly on purpose to retain focus. MC dropout is neither fast nor exciting, but does its job and unlikely
introduces complications that would raise suspicions/complications with the model. Extending beyond it is left as future
work. **(2)** In both Fig. 1–2 the number of hidden units is left small (also no ensembling or such used) to highlight the
noisiness of the corresponding NN models. In Fig. 1, the uncertainty does not always increase right outside of the data
range as it does for the GP models on the top. This property reflects the remaining suboptimality of the model. In Fig. 2,
the trained models end up in different local optima (this could probably be tuned). Mean-reversion is characteristic for
stationary models (outside data the model knows it's uncertain and reverts to the mean) and a desired property (also
applies to the RBF fig). **(3)** The activation func. lengthscale parameter is fixed in all the NN experiments, because the
preceding layer(s) take care of scaling the inputs, which serves the same purpose. We will discuss this in the main paper.
**(4)** As discussed in Sec. 5, the only practical problems were encountered with the spiky (non-differentiable) Matérn-1/2
activation. In general, tuning the learning rate might also help prevent possible issues with convergence.

[Meta-Review · NeurIPS 2020]

The authors proposes activation functions derived from stationary Matern family kernels which is widely used in Gaussian Process and can approximate uncertainty. Reviewers found that the paper to be well motivated and clearly described in the context of previous related work. This could be further improved by expanding the discussion to other GP kernels similar to Matern and the reason for the specific choice of Matern in that larger context. The empirical results were adequate but could be improved. The Fig 1 and 2 need further elaboration and analysis to explain the anomalies pointed out by Reviewer #4. There were additional concerns about experimental parameters and replicability. See comments from Reviewer #4.